# Switching of metabolic programs in response to light availability is an essential function of the cyanobacterial circadian output pathway

Anna M Puszynska[1,2,3]*, Erin K O'Shea[1,2,3,4]*

[1]Department of Molecular and Cellular Biology, Harvard University, Cambridge, United States; [2]Faculty of Arts and Sciences Center for Systems Biology, Harvard University, Cambridge, United States; [3]Howard Hughes Medical Institute, Harvard University, Cambridge, United States; [4]Department of Chemistry and Chemical Biology, Harvard University, Cambridge, United States

**Abstract** The transcription factor RpaA is the master regulator of circadian transcription in cyanobacteria, driving genome-wide oscillations in mRNA abundance. Deletion of *rpaA* has no effect on viability in constant light conditions, but renders cells inviable in cycling conditions when light and dark periods alternate. We investigated the mechanisms underlying this viability defect, and demonstrate that the *rpaA*⁻ strain cannot maintain appropriate energy status at night, does not accumulate carbon reserves during the day, and is defective in transcription of genes crucial for utilization of carbohydrate stores at night. Reconstruction of carbon utilization pathways combined with provision of an external carbon source restores energy charge and viability of the *rpaA*⁻ strain in light/dark cycling conditions. Our observations highlight how a circadian output pathway controls and temporally coordinates essential pathways in carbon metabolism to maximize fitness of cells facing periodic energy limitations.

*For correspondence: apuszyns@ fas.harvard.edu (AMP); erin_oshea@harvard.edu (EKO)

## Introduction

Organisms across kingdoms of life have evolved circadian clocks to temporally align biological activities with the diurnal changes in the environment. In the cyanobacterium *Synechococcus elongatus* PCC7942, the core circadian oscillator is comprised of the KaiA, KaiB and KaiC proteins (*Nishiwaki et al., 2007*; *Rust et al., 2007*) that generate oscillations in the phosphorylation state of KaiC. Time information encoded in the phosphorylation state of KaiC is transmitted to the transcription factor RpaA (*Takai et al., 2006*; *Taniguchi et al., 2010*; *Markson et al., 2013*) to generate circadian changes in gene expression (*Markson et al., 2013*). In continuous light, the expression of more than 60% of protein-coding genes in *S. elongatus* is regulated in a circadian manner (*Vijayan et al., 2009*) with two main phases of gene expression: genes peaking at subjective dawn or subjective dusk, where the term 'subjective' refers to an internal estimate of time in the absence of external cues. Deletion of *rpaA* disrupts these rhythms in mRNA abundance and arrests cells in a subjective dawn-like transcriptional state, rendering them unable to switch to a subjective dusk-like expression program (*Markson et al., 2013*).

While it is clear that the circadian transcriptional program schedules timing of gene expression when the clock is free-running in the absence of changes in external light, it is unclear how circadian control of gene expression is used under more physiologically relevant conditions when light and dark periods alternate. Exposure to darkness restricts energy availability, creating unique metabolic

**eLife digest** The cycle of day and night is one of the most recurrent and predictable environmental changes on our planet. Consequently, organisms have evolved mechanisms that allow them to measure time over 24 hours and prepare for the periodic changes between light and dark. These mechanisms, known as circadian clocks, alter the activity of some of the organism's genes in a rhythmic way across the course of a day. This in turn causes certain behaviors and biological activities of the organism to follow a daily cycle.

The bacterium *Synechococcus elongatus* needs to be able to track the daily cycle of light and dark because it performs photosynthesis and depends on sunlight to form sugars, which can later be broken down to release energy. The time information encoded in the circadian clock of *S. elongatus* is transmitted to the protein RpaA, which drives the regular circadian changes in gene activity in the cell. If RpaA is removed from the cell or prevented from working, *S. elongatus* can no longer control rhythmic gene activity and is unable to survive the night.

To better understand how the circadian system schedules biological tasks to help an organism to survive, Puszynska and O'Shea studied *S. elongatus* cells. This revealed that the bacteria normally prepare for darkness by storing sugars during the day and activating several genes at dusk to make enzymes that are required to break down stored sugars. This provides the cells with energy that they need to survive the night. But mutant cells that lack the gene that produces RpaA do not prepare for darkness; they do not accumulate a store of sugars during the day or activate the vital genes at dusk. They have low internal energy levels in the dark and they cannot survive long periods of darkness.

Providing the mutant cells with sugar and restoring the activity of the genes responsible for breaking down sugar enabled the cells to maintain energy in darkness and survive the night. It therefore appears that one role of the circadian system of *S. elongatus* is to coordinate building up sugar reserves during the day with breaking down sugar stores to generate energy during the night.

Puszynska and O'Shea also found many other genes that are not activated at dusk in the mutant cells. It will therefore be important to study whether other pathways that help cells to survive and grow are defective in these mutant cells.

demands in cyanobacteria which depend on sunlight for energy production through photosynthesis. Strikingly, the *rpaA*⁻ strain exhibits defects in cell growth and viability in cyclic, but not in constant light environments (*Takai et al., 2006*), suggesting an important role for circadian regulation of gene expression in alternating light/dark cycles.

We find that both accumulation and utilization of the carbon reserves required for energy production during periods of darkness are defective in the *rpaA*⁻ strain, and that correction of these defects restores viability. Our results provide insight into the role of the circadian output pathway in enhancing fitness of cyanobacteria through coordination of central carbon metabolism with the metabolic demands imposed by periodic changes in the external environment.

## Results

As reported previously (*Takai et al., 2006*), in constant light wild type and the *rpaA*⁻ cells grow at the same rate (*Figure 1A*, top panel); however, the *rpaA*⁻ strain is not viable when cultured in alternating light/dark conditions (*Figure 1A*, bottom panel and *Figure 1—figure supplement 1A*). The *rpaA*⁻ strain rapidly loses viability when incubated in the dark (*Figure 1—figure supplement 1B*), suggesting that the defect is induced by darkness and not by repeated light-to-dark and dark-to-light transitions. Complementation of the *rpaA*⁻ strain with *rpaA* expressed from an ectopic site in the genome fully restores viability (*Figure 1—figure supplement 1A* and *Figure 1—figure supplement 1B*). In wild type cells, expression of the *kaiBC* genes depends on RpaA and thus deletion of *rpaA* abrogates the function of the KaiABC oscillator, altering the stoichiometry of the core clock proteins and abrogating the rhythms in KaiC phosphorylation (*Takai et al., 2006*). To establish whether the defect in viability during dark periods results from loss of Kai oscillator function or from loss of RpaA function, we performed viability experiments using the 'clock rescue' strain background

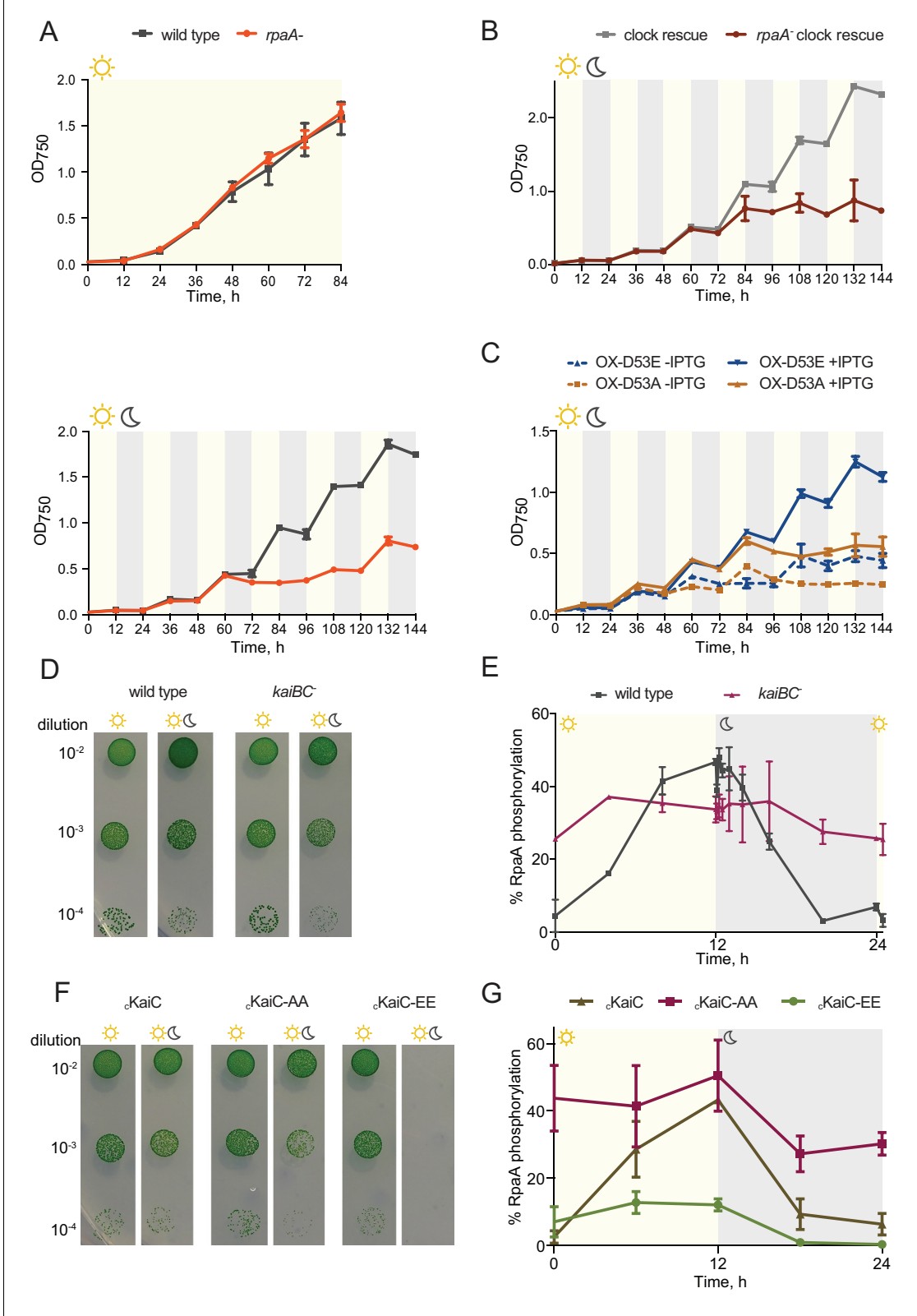

**Figure 1.** The *rpaA⁻* strain displays a defect in viability in light/dark conditions. (**A**) Growth curves of wild type and the *rpaA⁻* strain in constant light (top) and in 12 hr light/12 hr dark conditions (below) in BG-11 medium. Points represent the mean of three independent experiments with error bars displaying the standard error of the mean. (**B**) Growth curves of the 'clock rescue' and the *rpaA⁻* 'clock rescue' strains in 12 hr light/12 hr dark conditions in BG-11 medium. Points represent the mean of three independent experiments with error bars displaying the standard error of the mean. (**C**) Growth

*Figure 1 continued on next page*

*Figure 1 continued*

curves of the *rpaA⁻ kaiBC⁻* strains containing a P*trc* promoter driving expression of RpaA D53E (OX-D53E) or RpaA D53A (OX-D53A) in 12 hr light/12 hr dark conditions grown in the presence or absence of 20 µM IPTG. Points represent the mean of two biological replicates with error bars displaying the standard error of the mean. (D) Comparison of growth of wild type and the *kaiBC⁻* mutant in constant light and in light/dark conditions. The experiment was performed three independent times and a representative experiment is shown. (E) Quantification of RpaA phosphorylation levels measured by Phos-tag western blot in wild type and the *kaiBC⁻* strains in light/dark conditions. Each point represents the mean of two biological replicates with error bars displaying the standard error of the mean. (F) Comparison of growth of the ₍c₎KaiC strain (subscript 'c' denoting constitutive, RpaA-independent expression) and strains expressing phosphomimetic variants of KaiC, ₍c₎KaiC-EE or ₍c₎KaiC-AA, in constant light and in light/dark conditions. The experiment was performed three independent times and a representative experiment is shown. (G) Quantification of phosphorylation levels of RpaA measured by Phos-tag western blot in the ₍c₎KaiC strain or in the strains expressing phosphomimetic variants of KaiC, ₍c₎KaiC-EE and ₍c₎KaiC-AA, in light/dark conditions. Each point represents the mean of three biological replicates with error bars displaying the standard error of the mean.

The following figure supplements are available for figure 1:

**Figure supplement 1.** Characterization of growth and viability of the *rpaA* mutants.

**Figure supplement 2.** Characterization of protein levels of KaiC and RpaA in the ₍c₎KaiC phosphomimetic mutants in comparison to wild type.

in which *kaiBC* expression is made independent of RpaA (*Teng et al., 2013*). We observed that the *rpaA⁻* 'clock rescue' strain phenocopies the *rpaA⁻* strain and the isogenic *rpaA⁺* 'clock rescue' strain phenocopies wild type strain, demonstrating that the viability defect stems from the loss of RpaA function and not from loss of Kai oscillator function (*Figure 1B* and *Figure 1—figure supplement 1C*).

Since phosphorylation of RpaA is required for binding to its target promoters and gene expression (*Markson et al., 2013*), we asked whether the *rpaA* deletion phenotype in light/dark conditions can be rescued by a specific phosphorylated form of RpaA: RpaA D53E mimicking the phosphorylated state of RpaA; and RpaA D53A mimicking the unphosphorylated state. We examined the viability of the OX-D53E and OX-D53A strains, which lack both *kaiBC* and *rpaA* and contain a phosphomimetic form of RpaA expressed from an IPTG-inducible P*trc* promoter (*Markson et al., 2013*). In these strains RpaA phosphorylation is not controlled by either the clock activity or by any other potential input factor, isolating RpaA activity from its regulators. We observed that expression of the RpaA D53E phosphomimetic restores viability in light/dark conditions, while expression of the RpaA D53A variant does not (*Figure 1C*). We also compared growth of the ₍c₎RpaA strain (subscript 'c' denotes constitutive, RpaA-independent expression; see Materials and methods) with the ₍c₎RpaA D53A strain, each made using the 'clock rescue' background, in light/dark cycles. We found that while the control ₍c₎RpaA strain grows under these conditions despite lower levels of RpaA driven by the constitutive promoter, the ₍c₎RpaA D53A strain is inviable (*Figure 1—figure supplement 1D and E*). We conclude that the active DNA-binding form of RpaA is necessary and sufficient for cell viability during exposure to light/dark conditions.

We next analyzed the role of the core KaiABC clock in sustaining viability of cyanobacterial cells in light/dark conditions. Deletion of the *kaiBC* operon leads to only a minor viability defect in light/dark conditions (*Boyd et al., 2013* and *Figure 1D*) and substantial RpaA phosphorylation levels were observed in the *kaiBC⁻* strain at the subjective dawn (CT = 2 hr) and afternoon (CT = 9 hr) in constant light conditions (*Boyd et al., 2013*). Since RpaA phosphorylation levels are regulated by the antagonistic activities of two clock-controlled histidine kinases, SasA and CikA, and complexes of KaiB and KaiC proteins activate the phosphatase activity of CikA in vitro (*Gutu and O'Shea, 2013*), we hypothesized that RpaA would remain highly phosphorylated throughout the light/dark cycle in the *kaiBC⁻* strain. In agreement with this prediction, when we analyzed RpaA phosphorylation in the *kaiBC⁻* mutant we detected high levels of RpaA phosphorylation both in light and in the dark (*Figure 1E*). Thus, the ability of the *kaiBC⁻* mutant to survive alternating periods of light and dark likely stems from the high levels of phosphorylated RpaA in this strain. While the *kaiBC* operon is not essential for cell viability in light/dark cycles, specific KaiC phosphoforms, whose relative abundances vary during the circadian cycle (*Rust et al., 2007*), may have different effects on RpaA phosphorylation and, consequently, affect the ability of cells to survive light/dark cycling conditions. We tested this hypothesis by analyzing cell viability and RpaA phosphorylation in strains expressing KaiC

point mutants: KaiC-AA (KaiC S431A T432A), representing the unphosphorylated phosphoform of KaiC, and KaiC-EE (KaiC S431E T432E), representing the hyperphosphorylated KaiC variant. We used the P*0050* promoter, which releases KaiC and RpaA expression from circadian feedback, to ensure equal expression of KaiC variants as well as of RpaA between the phosphomimetic strains (*Figure 1—figure supplement 2*). We observed that while the _c_KaiC-AA mutant is viable in light/ dark conditions, the _c_KaiC-EE strain is not able to grow (*Figure 1F*). Strikingly, the _c_KaiC-AA strain maintains high RpaA phosphorylation levels during the diurnal cycle, while RpaA is phosphorylated in the _c_KaiC-EE strain only at a low level (*Figure 1G*). This ability to control RpaA phosphorylation and cell viability through perturbations in KaiC phosphorylation status underscores the primacy of this regulatory axis in the diurnal control of RpaA phosphorylation, and suggests that any hypothetical clock-independent environmental factors play a lesser role in regulating RpaA phosphorylation under the conditions we tested.

Circadian clocks and their output synchronize the metabolic activity of cells with diurnal fluctuations (*Tu and McKnight, 2006*; *Green et al., 2008*; *Graf et al., 2010*; *Dodd et al., 2015*). To understand if the observed viability defect stems from an inability of the *rpaA⁻* strain (which is arrested in a 'dawn-like' transcriptional state [*Markson et al., 2013*]) to meet the metabolic requirements of darkness, we assessed cellular energy charge in wild type and the *rpaA⁻* strain over time in constant light conditions and in light/dark cycles. Energy charge describes the relative levels of ATP, ADP and AMP in the intracellular adenine nucleotide pool, and is a tightly controlled parameter that reflects the global metabolic energy status of the cell (*Atkinson, 1968*). In constant light, there was no difference in energy charge between wild type and the *rpaA⁻* strain (*Figure 2*, top panel). Wild type cells subjected to darkness experience only a slight reduction in energy charge (*Figure 2*, bottom panel, *Figure 2—figure supplement 1A*). In contrast, *rpaA⁻* cells experience a reduction in energy charge during the first dark period and an even more dramatic reduction during the second dark period, suggesting that this strain has a defect in energy metabolism affecting energy maintenance during the dark phase of the light/ dark cycle (*Figure 2*, bottom panel, *Figure 2— figure supplement 1B*). The decrease in energy charge was accompanied by a reduction in the total adenine nucleotide pool size in the *rpaA⁻* strain (*Figure 2—figure supplement 2*). A similar trend has been previously observed during starvation in carbon-limited cultures of *E. coli* (*Chapman et al., 1971*), prompting us to analyze changes in activity through pathways relevant for carbon metabolism in the strain lacking *rpaA*.

In constant light conditions, the circadian program directs expression of anabolic and catabolic pathways, with genes relevant for carbon catabolism peaking in mRNA abundance at

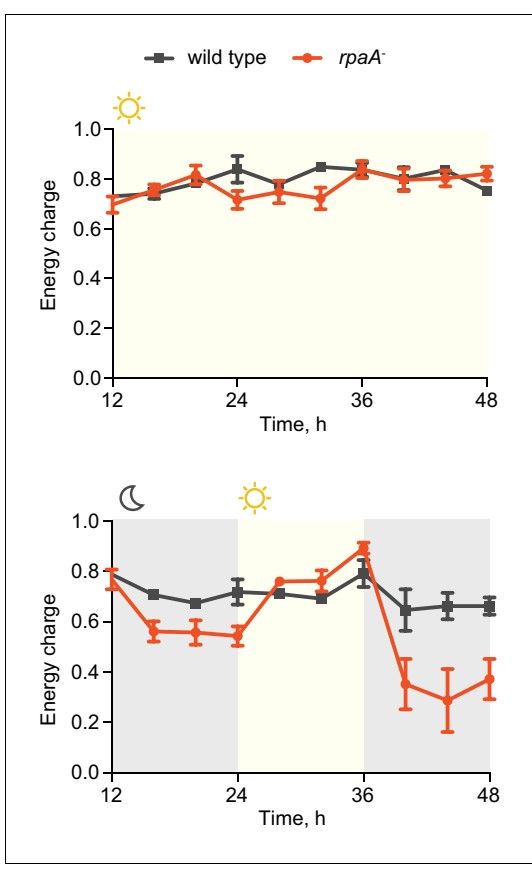

**Figure 2.** The *rpaA⁻* strain has a defect in maintenance of the cellular energy charge in oscillating light/dark conditions. Changes in the energy charge, defined as ([ATP] +0.5*[ADP])/([ATP]+[ADP]+[AMP]), during growth of wild type and the *rpaA⁻* strain in constant light (top) or in 12 hr light/12 hr dark conditions (bottom) in BG-11 medium. Each point represents the mean of three independent experiments with error bars displaying the standard error of the mean.

The following figure supplements are available for figure 2:

**Figure supplement 1.** Changes in the ATP, ADP and AMP levels in cells grown in light/dark conditions.

**Figure supplement 2.** The *rpaA⁻* strain displays a defect in levels of adenine nucleotides in light/dark cycles.

subjective dusk (*Vijayan et al., 2009*; *Diamond et al., 2015*). To investigate the role of RpaA in orchestrating diurnal transcription of metabolic genes, we performed RNA sequencing in the *rpaA*⁻ 'clock rescue' strain and an isogenic *rpaA*⁺ strain after exposure to darkness. The expression of circadian dusk-peaking genes was most affected in the *rpaA* deletion strain exposed to darkness (*Figure 3—figure supplement 1A*). Specifically, we observed that genes encoding enzymes involved in glycogen breakdown, glycolysis and the oxidative pentose phosphate pathway (*Figure 3—figure supplement 1B* and *Supplementary file 1*), which normally peak in expression at dusk, are among the genes with the greatest defect in expression in the *rpaA*⁻ cells (*Figure 3A*). These pathways are key for carbon utilization and energy production in the dark in cyanobacteria (*Figure 3B*), and their function is essential for survival of periods of darkness (*Doolittle and Singer, 1974*). Independent deletion of *gnd* as well as the *glgP_gap1* and *fbp_zwf_opcA* operons, which exhibit severely reduced expression in the *rpaA*⁻ strain, results in impaired viability in light/dark cycles but not in constant light conditions (*Figure 3—figure supplement 1C* and *Doolittle and Singer, 1974*; *Scanlan et al., 1995*). We also measured expression of the key dusk metabolic genes, *gnd*, *glgP* and *zwf*, in the *kaiBC*⁻ strain at dusk and in the first hours of darkness (*Figure 3—figure supplement 2*). We observed that these genes are expressed at a high level, further emphasizing the role of phosphorylated RpaA in activation of the dusk transcriptional program and maintenance of cell viability in light/dark conditions. To confirm that the transcriptional defects in the *rpaA* deletion mutant affect carbon catabolism, we measured the enzymatic activities of glycogen phosphorylase (*glgP*), glucose-6-phosphate dehydrogenase (*zwf*) and 6-phosphogluconate dehydrogenase (*gnd*) in the dark, and observed that activity of each enzyme was strongly reduced in the *rpaA*⁻ strain compared to wild type (*Figure 3C*). Therefore, the *rpaA*⁻ strain appears to be unable to activate the carbon catabolic pathways upon the onset of darkness because it cannot induce transcription of the requisite enzymes at dusk.

Glycogen is a crucial metabolic reserve used as a carbon and energy source during periods of darkness (*Smith, 1983*). In wild type cyanobacteria glycogen accumulates during the afternoon (*Cervený and Nedbal, 2009*; *Knoop et al., 2010*; *Rügen et al., 2015*) through the activity of a glycogen synthesis pathway composed of phosphoglucomutase (*pgm1* and *pgm2*), ADP-glucose pyrophosphorylase (*glgC*) and glycogen synthase (*glgA*) (*Figure 4A*). To analyze whether the dawn-arrested *rpaA*⁻ strain can accumulate glycogen reserves, we measured expression and activity of the glycogen synthesis enzymes and found that neither the expression levels nor activity of the enzymes in the pathway were significantly reduced in the *rpaA*⁻ strain (*Figure 4B and C*). Surprisingly, we observed that while glycogen content oscillates over a period of 24 hr in wild type cells both in constant light and in light/dark cycles, glycogen is present at a very low level in the *rpaA*⁻ strain regardless of light conditions, despite high levels of the synthetic enzymes in the extracts (*Figure 4D*). The same glycogen accumulation defect occurs in the *rpaA*⁻ 'clock rescue' strain, demonstrating that it is a direct effect of deletion of *rpaA* (*Figure 4E*). Like the *rpaA*⁻ strain, glycogen synthesis mutants (*glgA*⁻ and *glgC*⁻) exhibit reduced viability in alternating light/dark conditions (*Gründel et al., 2012*), *Figure 4—figure supplement 1A*, bottom panel) and in constant darkness (*Figure 4—figure supplement 1B*), but grow somewhat slower than the *rpaA*⁻ strain in constant light conditions (*Figure 4—figure supplement 1A*, top panel). We conclude that the deficiency in the preparation of a reserve carbon source – which occurs not at the level of transcription of the glycogen synthesis genes or regulation of the activity of the encoded enzymes – may contribute to the impaired viability of the *rpaA*⁻ strain in the dark.

We hypothesized that the inability of the *rpaA*⁻ strain to maintain appropriate energy levels and cell viability in the dark stems from the defects in accumulation of carbon/energy stores during the day and their utilization at night. To assess whether restoration of the correct carbon catabolic route in darkness is sufficient to rescue the energy charge and viability of the *rpaA*⁻ strain in light/dark conditions, we reconstituted the activities of the carbon utilization pathways and provided a transporter for uptake of external carbon. We restored expression of the carbon utilization enzymes in the *rpaA*⁻ strain by placing their expression under an IPTG-inducible ectopic P*trc* promoter, and confirmed that induction with IPTG restored enzyme activity in cell extracts (*Figure 5—figure supplement 1*). To counterbalance the inability of the *rpaA*⁻ cells to accumulate internal carbon reserves, we expressed the GalP transporter to allow glucose uptake from the medium. Expression of either the missing enzymes or the GalP transporter alone with supplementation of glucose in the *rpaA*⁻ background was not sufficient to restore viability of this strain (*Figure 5A and B*). However, restoration of

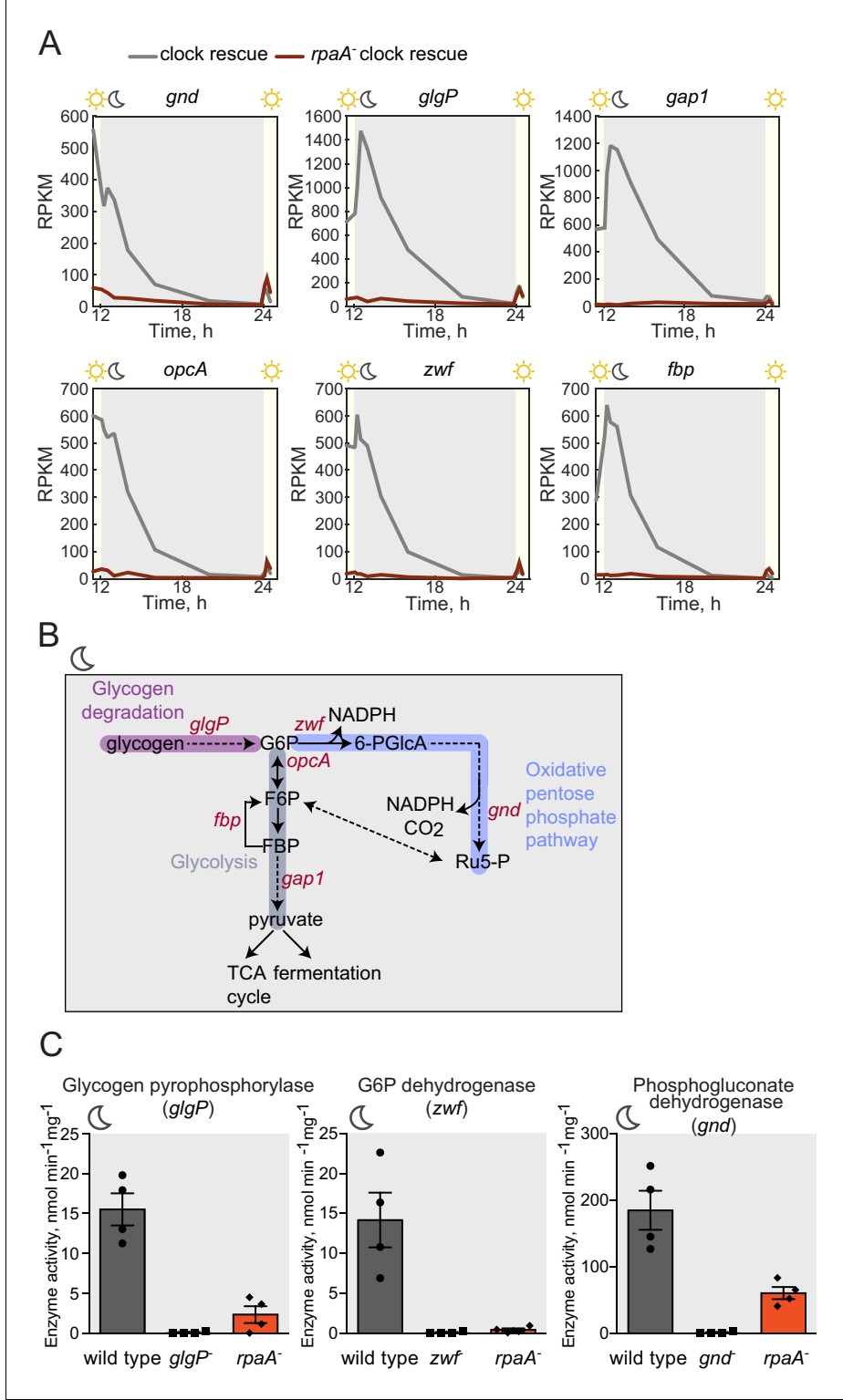

**Figure 3.** Sugar catabolism pathways are abrogated in the *rpaA⁻* strain. (**A**) Temporal expression profiles of genes encoding enzymes involved in sugar metabolism whose expression is abrogated in the *rpaA⁻* 'clock rescue' strain. Relative RNA levels were measured in the light/dark conditions by RNA sequencing. (**B**) A diagram representing carbon metabolism in *S. elongatus* PCC7942 in the dark. At night glycogen constitutes the store for energy and carbon skeletons. Glycogen degradation, glycolysis and oxidative pentose phosphate pathways are the key metabolic pathways operating at night. Deletion of operons with genes colored red leads to a strong viability

*Figure 3 continued on next page*

*Figure 3 continued*

defect specifically in light/dark conditions (**Figure 3—figure supplement 1C** and **Doolittle and Singer 1974**; **Scanlan et al., 1995**). *glgP*, glycogen phosphorylase; *gap1*, glyceraldehyde-3-phosphate dehydrogenase; *gnd*, 6-phosphogluconate dehydrogenase; *zwf*, glucose-6-phosphate dehydrogenase; *opcA*, glucose-6-phosphate dehydrogenase assembly protein; *fbp*, fructose-1,6-bisphosphatase, G6P, glucose-6-phosphate; 6-PGlcA, 6-phosphogluconate; F6P, fructose-6-phosphate; FBP, fructose-1,6-bisphosphate, Ru5-P, ribulose-5-phosphate (C) Enzymatic activities of glycogen phosphorylase, glucose-6-phosphate dehydrogenase and 6-phosphogluconate dehydrogenase in wild type, the *rpaA*⁻ and negative control strains measured 3 hr after exposure to expected darkness. Error bars represent standard error of the mean of four independent experiments.

The following figure supplements are available for figure 3:

**Figure supplement 1.** Expression of genes encoding enzymes involved in alternative sugar catabolism pathways is defective in the *rpaA*⁻ strain in light/dark cycles.

**Figure supplement 2.** Expression of key metabolic dusk genes in the *kaiBC*⁻ strain.

---

the carbon utilization pathways combined with the introduction of the glucose transporter in the presence of glucose rescued the viability defect of the *rpaA*⁻ strain (**Figure 5A and B**). Glucose feeding does not increase accumulation of internal carbon stores in this engineered strain (**Figure 5C**), nor in wild type expressing the GalP transporter (**Figure 5—figure supplement 2**). However, it is sufficient to restore high energy charge in the *rpaA*⁻ strain expressing both the key catabolic enzymes and GalP (**Figure 5D**). Our results strongly suggest that functional circadian output mediated by the activity of RpaA is required for the anticipatory accumulation of carbon reserves during the day, as well as metabolic switching to carbon catabolism at dusk. These metabolic adaptations allow cells to stimulate energy production and sustain cell viability during periods of darkness that challenge and constrain resources for cyanobacterial growth.

## Discussion

Daily environmental changes in light availability give rise to periodic demand for metabolism of alternate energy sources in photosynthetic organisms (**Doolittle, 1979**). To maximize fitness, photosynthetic organisms need to accurately allocate resources by carrying out carbon assimilation during the day and utilizing stored carbohydrate reserves at night (**Smith, 1983**). Here we show that the phosphorylated form of the transcription factor RpaA, regulated by the circadian clock, is critical for the coordination of carbon anabolism and catabolism and thus cell viability in light/dark cycles. Deletion of *rpaA* prevents glycogen accumulation in light and renders cells unable to utilize existing glycogen at dusk (**Figure 5E**) – this leads to alterations in cellular energy charge and, as previously observed, to a severe reduction in fitness. It is yet to be determined whether RpaA is rewiring carbon metabolism solely through transcriptional control or through other mechanisms.

The mechanism underlying perturbed carbon accumulation in the *rpaA*⁻ strain is unclear. The activities of glycogen synthesizing enzymes in the *rpaA*⁻ strain are high in vitro, however they could be affected in vivo by the redox state of the cell or by metabolite levels. It is also possible that an RpaA-dependent transcript expressed in the afternoon is required to funnel assimilated carbon into glycogen synthesis. Alternatively, additional metabolic flux changes may occur in the *rpaA*⁻ strain that prevent formation of glycogen stores. Finally, it is formally possible that the *rpaA*⁻ cells perform a futile cycle, simultaneously synthesizing and breaking glycogen down. However, since the activity of the glycogen phosphorylase involved in glycogen degradation is low in the *rpaA*⁻ strain, we find this scenario less likely.

Our data support a model in which anticipation of periodic dark-induced resource limitation is one essential role of the cyanobacterial circadian system. The output of the core oscillator, through regulation of RpaA activity, temporally coordinates metabolism to prepare cells for the metabolic demands of periods of darkness. Correct scheduling of central carbon metabolic activities allows cells to survive the night. Recently, Lambert et al. observed that the circadian clock orchestrates a trade-off between rapid cell growth in the morning and robustness to starvation in the afternoon,

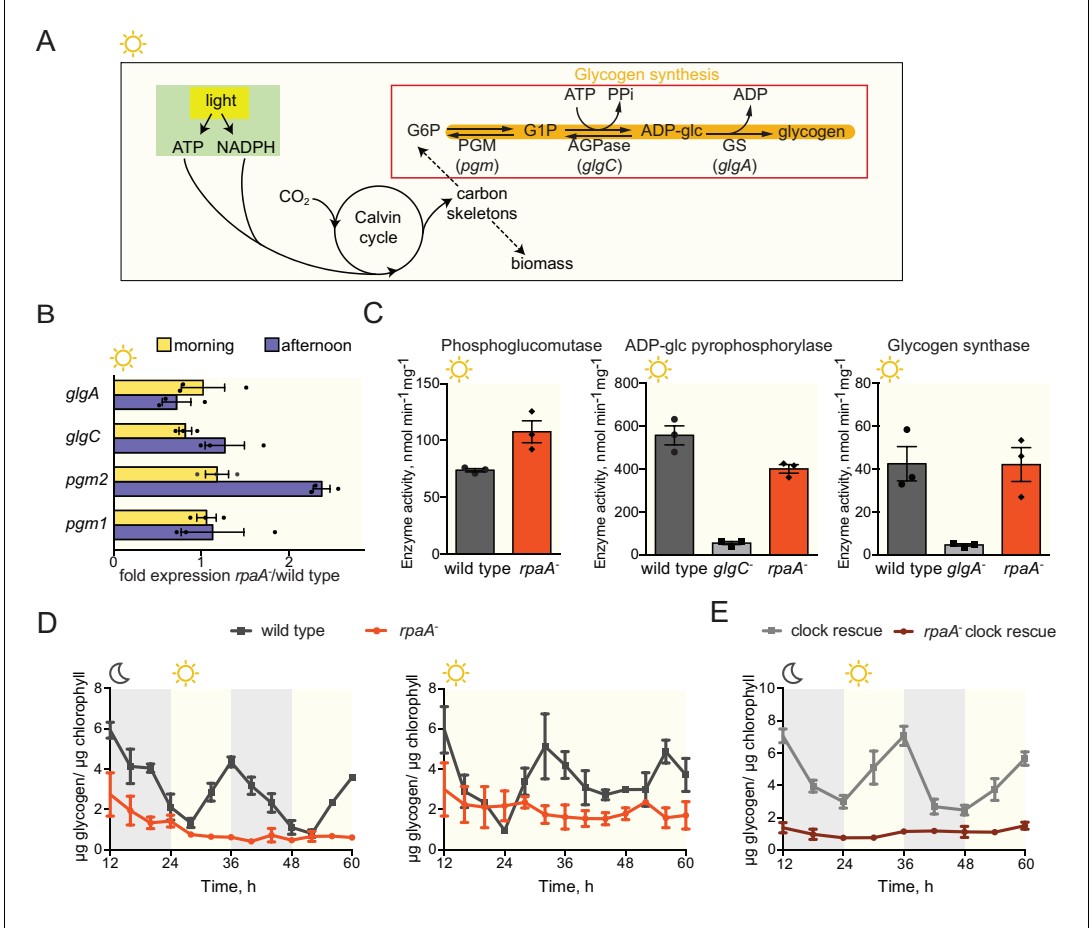

**Figure 4.** The *rpaA*⁻ strain accumulates little glycogen. (**A**) A diagram representing carbon metabolism in *S. elongatus* PCC7942 in light. During the day *S. elongatus* cells perform photosynthesis to produce carbon skeletons and energy for growth and for preparation of glycogen stores. The glycogen synthesis pathway is comprised of three enzymatic activities: phosphoglucomutase (PGM), ADP-glucose pyrophosphorylase (AGPase) and glycogen synthase (GS). G6P, glucose-6-phosphate; G1P, glucose-1-phosphate; *pgm*, phosphoglucomutase; *glgC*, ADP-glucose pyrophosphorylase; *glgA*, glycogen synthase. (**B**) Relative expression of genes encoding enzymes in the glycogen synthesis pathway measured by RT-qPCR in the morning and in the afternoon in wild type and the *rpaA*⁻ strains. Error bars represent the standard error of the mean of three independent experiments. (**C**) Activities of enzymes in the glycogen synthesis pathway in wild type, the *rpaA*⁻ and negative control strains measured during the day (at time = 10 hr). Error bars represent the standard error of the mean of three independent experiments. (**D**) Glycogen content in wild type and the *rpaA*⁻ strains grown in light/dark and constant light conditions. Points represent the mean of two experiments with error bars displaying the standard error of the mean. (**E**) Glycogen content in the 'clock rescue' and the *rpaA*⁻ 'clock rescue' strains grown in light/dark conditions. Points represent the mean of two experiments with error bars displaying the standard error of the mean.

The following figure supplement is available for figure 4:

**Figure supplement 1.** Comparison of growth and viability of the *rpaA*⁻ strain and glycogen synthesis mutants.

suggesting preparation for the night as a plausible role for the circadian clock (*Lambert et al., 2016*). Our findings are consistent with this model and provide novel mechanistic insight into circadian regulation of cyanobacterial energy metabolism. It remains to be investigated whether the circadian-driven preparation for darkness is a shared strategy employed by other photosynthetic organisms.

Further, our results suggest a molecular explanation for the observed selective advantage of circadian resonance. When the periods of the endogenous circadian system and periods of the environmental fluctuations are equal, organisms show a fitness advantage and outcompete mutants whose clocks have an altered periodicity in the same environment (*Ouyang et al., 1998*;

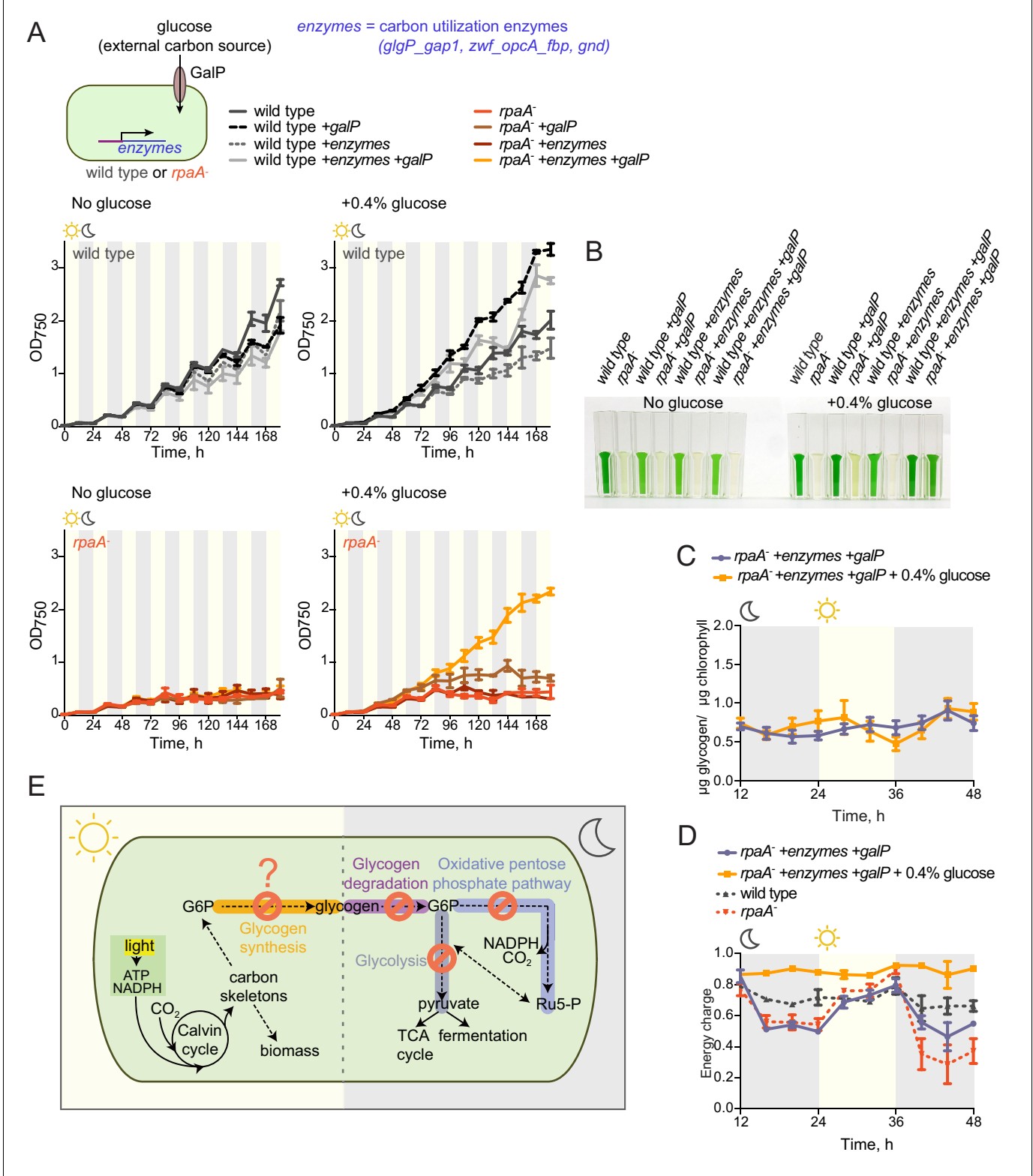

**Figure 5.** Restoration of sugar catabolism pathways and glucose feeding rescue the viability defect of the *rpaA*⁻ strain. (**A**) Growth curves of the indicated engineered strains in 12 hr light/12 hr dark conditions in BG-11 medium with 100 μM IPTG and with (right) or without (left) 0.4% glucose supplementation. Wild type (top) and the *rpaA*⁻ (below) strains were engineered to express a sugar transporter GalP and/or sugar utilization enzymes using an IPTG-inducible promoter. Points represent the mean of two experiments with error bars displaying the standard error of the mean. (**B**) Representative cell cultures of the engineered strains photographed at the end of the growth experiment in light/dark conditions with or without 0.4%

*Figure 5 continued on next page*

*Figure 5 continued*

glucose supplementation. *S. elongatus* cells are blue-green in color. The darker the color of the culture, the higher is its optical density at 750 nm. (**C**) Glycogen content in in the *rpaA⁻ + enzymes + galP* strain ±0.4% glucose in light/dark conditions. Points represent the mean of two experiments with error bars displaying the standard error of the mean. (**D**) Energy charge measurement in the *rpaA⁻ + enzymes + galP* strain with or without 0.4% glucose supplementation in light/dark conditions. Points represent the mean of two experiments with error bars displaying the standard error of the mean. Data representing the energy charge in wild type and the *rpaA⁻* strain are reproduced from *Figure 2* to facilitate comparison. (**E**) A model representing physiological processes in *S. elongatus* regulated by the activity of the circadian program that contribute to cell fitness. The circadian clock schedules periods of activity of RpaA to coordinate anticipatory carbon reserve formation during the day with carbon utilizing metabolic pathways activated at dusk to fuel cell integrity and viability in an environment in which light and dark periods alternate. Defects present in the *rpaA* deficient cells, indicated by red symbols on the diagram, lead to a low energy charge and reduced viability during periods of darkness.

The following figure supplements are available for figure 5:

**Figure supplement 1.** Characterization of strains used for the reconstitution experiments.

**Figure supplement 2.** Glucose supplementation does not increase glycogen accumulation in wild type cells.

---

*Woelfle et al., 2004*). The mechanistic basis for this observation has been lacking. Based on our findings, we expect that the fitness defects in the strains with core oscillator periods unequal to the period of environmental variation result from mistiming of RpaA activation. The resultant mistiming of accumulation and mobilization of glycogen stores with the onset of the nightfall would lead to a carbon deficit at night, and impact growth as in the *rpaA⁻* mutant during dark periods. One role of the KaiABC clock in our model, therefore, is to correctly phase the activity of RpaA and downstream carbon metabolism with the external environment, maximizing fitness and growth.

Though we show that RpaA activity is crucial for regulation of key aspects of carbohydrate metabolism, we identify ~90 genes whose expression is abrogated at dusk and throughout darkness in the *rpaA⁻* strain and it is possible that additional physiological processes are defective in the absence of *rpaA*. Moreover, although our results indicate that the KaiABC clock is the main regulator of diurnal RpaA phosphorylation, other factors may influence the activity of RpaA or its regulon under fluctuating environmental conditions. RpaA is known to cooperate with RpaB (*Espinosa et al., 2015*), a transcription factor crucial for responses to high light conditions. CikA, the clock-controlled regulator of RpaA phosphorylation, has been proposed to directly sense redox state of the cell (*Ivleva et al., 2006*). Finally, additional RpaA regulators, LabA and CikB, have been proposed (*Taniguchi et al., 2010*; *Boyd et al., 2016*). Hence, it is possible that in a rapidly changing environment RpaA integrates circadian cues with environmental inputs important for viability. It will be interesting to analyze how deletion of *rpaA* affects the ability of cells to respond to other environmental challenges that cyanobacteria routinely face.

Finally, our data together with a recent study by Diamond and colleagues (*Diamond et al., 2015*) emphasize the role of the circadian output pathway in driving oscillations in metabolism. On the other hand, it has been demonstrated that metabolic processes provide feedback to the circadian clock by regulating its resetting, indicating an important role of the fluctuating metabolic status of a cell in driving circadian oscillations (*Rust et al., 2011*; *Pattanayak et al., 2014*, *2015*). Together, a complex bidirectional relationship between the circadian program and metabolism emerges in which the circadian system both controls and is controlled by the metabolic rhythms to set and tune cell physiology with the environmental cycles. This tight coupling between circadian machinery and metabolism is coming to light as a universal feature across kingdoms of life (*Green et al., 2008*; *Dodd et al., 2015*).

## Materials and methods

### Cyanobacterial strains

Wild type *Synechococcus elongatus* PCC7942 was acquired from ATCC (RRID:SCR_001672) catalog no. 33912. Strains were constructed using standard protocols for genomic integration by homologous recombination (*Clerico et al., 2007*) and are listed in *Supplementary file 2*. All plasmids for strain construction were generated using Gibson assembly (*Gibson et al., 2009*) and verified by

Sanger sequencing. All strains were analyzed using colony PCR to verify target integration into the genome. Strains with point mutations introduced in the *kaiC* or *rpaA* open reading frames were additionally verified by PCR amplifying and Sanger sequencing of the relevant gene.

The *rpaA*⁻ strain was constructed by transforming wild type cells with pΔ*rpaA*(Gmr) that was a gift from Joe Markson. The 'clock rescue' strain was constructed following *Teng et al. (2013)* by replacing the native P*kaiBC* promoter between base pairs 1240468 and 1240554 on *S. elongatus* chromosome with a P*0050* promoter encompassing 508 base pairs upstream of the translation start of *synpcc7942_0050*. We used a kanamycin resistance cassette upstream of the promoter as a selection marker. The *rpaA*⁻ 'clock rescue' strain was made by transforming pΔ*rpaA*(Gmr) plasmid into the 'clock rescue' strain background. The *rpaA*⁻ +*rpaA* strain was made by transforming wild type *S. elongatus* simultaneously with pΔ*rpaA*(Gmr) and an NS1 targeting vector pAM1303 carrying the *rpaA* gene with its native promoter encompassing 400 base pairs upstream of the translation start. The pAM1303 plasmid was a gift from Susan Golden. The *rpaA*⁻ + *empty plasmid* was created by transforming wild type cells with a pΔ*rpaA*(Gmr) plasmid and an empty NS1 targeting vector pAM1303. The ꜀RpaA 'clock rescue' and ꜀RpaA D53A 'clock rescue' strains were both made using the 'clock rescue' strain background. In these strains the *rpaA* promoter was changed to a P*0050* promoter allowing for continuous expression of RpaA and RpaA D53A. A gentamycin resistance cassette was placed upstream of the P*0050* promoter for selection. The D53A mutation was introduced following *Markson et al. (2013)*.

The *kaiBC*⁻ strain was constructed by transforming wild type cells with pΔ*kaiBC*(Cmr) that was a gift from Joe Markson. To construct strains expressing phosphomimetic mutants of KaiC we transformed *kaiBC*⁻ cells with an NS1 targeting vector pAM1303 carrying *kaiBC* S431E T432E, *kaiBC* S431A T432A or *kaiBC* expressed from the P*0050* promoter allowing for constitutive expression of *kaiBC* variants. Because RpaA regulates its own expression, we replaced the *rpaA* promoter with the P*0050* promoter free of circadian feedback, ensuring equal and unperturbed expression of RpaA in the KaiC phosphomimetic mutants.

The strain lacking the *glgP_gap1* (*synpcc7942_0244–0245*) operon was made using the pBR322 plasmid carrying the kanamycin resistance cassette flanked by 1000 nucleotides of DNA from upstream and downstream of *glgP_gap1* locus. In the *zwf*⁻*fbp*⁻*opcA*⁻ strain the *zwf_fbp_opcA* operon (*synpcc7942_2333–2335*) was replaced with a kanamycin resistance cassette. In the *gnd*⁻ strain the *gnd* gene (*synpcc7942_0039*) was replaced with a kanamycin resistance cassette. The strain lacking *glgA* (*synpcc7942_2518*) and the strain lacking *glgC* (*synpcc7942_0603*) were made by using the pBR322 plasmids carrying the kanamycin resistance cassette flanked by 1000 nucleotides of DNA from upstream and downstream of respectively *glgA* and *glgC* locus.

The *wild type +galP* strain was made by transforming wild type cells with P*trc::galP* construct in a modified NS1 targeting vector pAM1303 in which the cassette providing resistance to spectinomycin and streptomycin was replaced with a nourseothricin resistance cassette (*Taton et al., 2014*). The gene *galP* was amplified from the *E. coli* genomic DNA. The *wild type +enzymes* strain was made by sequentially replacing P*gnd*, P*glg_gap1* and P*zwf_fbp_opcA* with the P*trc* promoter making expression of *gnd*, *glgP_gap1* and *zwf_fbp_opcA* transcripts IPTG-inducible. The *wild type +enzymes +galP* strain was made by transforming the *wild type +enzymes* strain with P*trc::galP* construct in a modified NS1 targeting vector pAM1303 that carried a nourseothricin resistance cassette. The *rpaA*⁻ +*galP*, *rpaA*⁻ +*enzymes* and *rpaA*⁻ +*enzymes +galP* strains were made by transforming pΔ*rpaA*(Gmr) plasmid respectively into *wild type +galP*, *wild type +enzymes* and *wild type +enzymes +galP* strains.

## Cell culture

Cell cultures of wild type and mutant cells were grown under illumination with cool fluorescent light at 40 µE m⁻² s⁻¹ (µmoles photons m⁻² s⁻¹) in BG11 medium at 30°C. For the light/dark experiments, cultures were incubated under alternating 12 hr light/12 hr dark conditions with the same light intensity of 40 µE m⁻² s⁻¹ during light periods. For the experiment performed in *Figure 1C* cells were grown in the presence or in the absence of 20 µM IPTG. For the experiments performed in *Figure 5* strains were grown in BG-11 medium with 100 µM IPTG with or without supplementation with 0.4% (w/v) glucose.

For the experiments described in *Figure 2* and *Figure 2—figure supplements 1* and *2*, *Figure 3*, *Figure 4*, *Figure 5—figure supplement 1* and *Figure 5—figure supplement 2* wild type or 'clock

rescue' strains were entrained by exposure to 12 hr of darkness, followed by 12 hr of light, followed by another 12 hr of darkness. The rpaA⁻ or rpaA⁻ 'clock rescue' strains in these experiments were entrained by one pulse of 12 hr of darkness, followed by incubation in light to allow cell growth to resume. For the experiments described in *Figure 1E*, *Figure 1G*, *Figure 3—figure supplement 2* cells were entrained by two pulses of 12 hr of darkness. For the experiments described in *Figure 5C and D* the strains were entrained by one pulse of 12 hr of darkness and then incubated in light until the start of the experiment.

## Growth and viability assays

For the liquid growth assays, liquid cultures of wild type and mutant cells were pre-grown in continuous light in a medium lacking antibiotics. Cultures were diluted to $OD_{750}$ = 0.02 and grown either in constant light or in 12 hr light/ 12 hr dark cycling conditions at 30°C. Optical density of cells was monitored at 750 nm. Experiments were performed in duplicate (*Figure 1C*, *Figure 4—figure supplement 1A*, *Figure 5A and B*) or in triplicate (*Figure 1A and B*).

For the spot plate growth assays, liquid cultures of wild type and mutant cells were pre-grown in continuous light in a medium lacking antibiotics. Cells were diluted to $OD_{750}$ = 0.25 and a dilution series was performed from $10^0$ to $10^{-4}$. Then, 10 µl of each dilution step were spotted onto BG11 agar plates with no antibiotics. Plates were incubated in constant light at 30°C for 7 days or under 12 hr light/ 12 hr dark alternating conditions for 14 days.

Viability after prolonged dark treatment was assessed by a colony forming unit assay. Light-grown cultures were diluted to $OD_{750}$ = 0.025, transferred to darkness and sampled every 24 hr. Each aliquot removed from the culture was diluted in BG11 medium 1000 fold by serial dilution. 100 µl of the diluted culture were plated onto a BG11 plate. Plates were incubated for 7 days at 30°C under constant illumination and colonies on each plate were counted. Viability was expressed as: % viability = $N/N_0 \times 100\%$, where $N_0$ is the colony count before exposure of cultures to darkness. Each experiment was performed in duplicate.

## Western blot analysis

Cells were harvested for western blotting by filtration and frozen in liquid nitrogen. To prepare protein samples, cells were resuspended in ice-cold lysis buffer (8 M urea, 20 mM HEPES, pH 8.0 and 1 mM β-mercaptoethanol) and ruptured by bead-beating with 0.1 mm glass beads for total of 5 min with periodic cooling on ice. Cell lysates were centrifuged for 12 min at 20,000 g at 4°C. Protein concentration of each sample was determined by Bradford assay (Bio-Rad). For experiment performed in *Figure 1—figure supplement 1E* and *Figure 1—figure supplement 2* lysates were loaded on 4– 20% Tris-Glycine Gel (Novex). Proteins were transferred to a nitrocellulose membrane using a semi-dry transfer apparatus (Bio-Rad). The membrane was blocked in TBST +2.5% milk and then incubated with an affinity-purified rabbit polyclonal antibody against RpaA or against KaiC (Cocalico Biologicals). To quantify phosphorylated RpaA in experiments presented in *Figure 1E* and *Figure 1G*, we performed Phos-tag western blots as described in *Markson et al. (2013)*. Quantification of western blots was performed using ImageJ software (RRID:SCR_003070).

## Adenine nucleotide analysis

Nucleotides were extracted following the method used by *Rust et al. (2011)* with modifications. 2 ml of the cyanobacterial culture were added to 0.5 ml of ice-cold 3 M perchloric acid with 77 mM EDTA, vortexed briefly and incubated on ice for 5 min. The mixture was neutralized with 1.34 ml of 1 M KOH, 0.5 M Tris, 0.5 M KCl and then centrifuged at 4000 rpm for 20 min at 4°C. The supernatant was then filtered through Amicon Ultra-4 filters (Millipore) and stored at –80°C.

To measure adenine nucleotides, extracts were thawed and diluted 2.6X. For ATP measurement, extracts were diluted in a buffer containing 25 mM KCl, 50 mM $MgSO_4$ and 100 mM HEPES, pH 7.4 (L buffer) with 1 mM phosphoenolpyruvate. To measure ADP + ATP, extracts were diluted in the above buffer containing also 3 U ml$^{-1}$ type II pyruvate kinase from rabbit muscle (Sigma-Aldrich). To measure AMP + ADP + ATP, extracts were diluted in the L buffer containing 1 mM phosphoenolpyruvate, 3 U ml$^{-1}$ type II pyruvate kinase from rabbit muscle and 75 U ml$^{-1}$ myokinase from rabbit muscle (Sigma-Aldrich). Diluted extracts were incubated for 30 min in a 37°C water bath followed by 10 min of heat treatment at 90°C to inactivate enzymes. Extracts were assayed in triplicate in a black

96-well plate. 30 μl of the L buffer containing 35 μg ml$^{-1}$ firefly luciferase (Sigma-Aldrich) and 1 mM luciferin (Sigma-Aldrich) were added to 260 μl of the extract. The luminescence signal was measured from each well in a TopCount luminescence counter (Perkin-Elmer).

## RNA-Seq

For the RNA-sequencing experiment the 'clock rescue' and the *rpaA*⁻ 'clock rescue' strains were entrained and harvested by filtering and freezing in liquid-nitrogen at following time points: dusk (ZT = 12 hr); 5 min, 15 min, 30 min, 1 hr, 2 hr, 4 hr, 8 hr and 11 hr 50 min after exposure to darkness; and 5 min, 15 min and 30 min after re-exposure to light at dawn.

RNA purification and library preparation were performed as described in *Markson et al. (2013)*. NCBI reference sequences NC_007604.1, NC_004073.2, and NC_004990.1 were used to align sequencing reads to the *S. elongatus* chromosome and the endogenous plasmids with Bowtie (RRID:SCR_005476). Uniquely mappable reads with maximum of three mismatches per read were allowed to map to the genome.

To quantify gene expression we counted the number of coding reads between the start and stop positions of open reading frames. We performed RPKM normalization and searched for differentially expressed genes that are at least 3-fold lower in the *rpaA*⁻ 'clock rescue' strain than in the control in at least 5 of the 12 measured time points. We performed an analysis of the functional annotations of the protein coding genes whose expression is defective in the *rpaA*⁻ 'clock rescue' strain using gene functions available in the Cyanobase (RRID:SCR_007615). The sequencing data reported in this article can be found in the NCBI GEO DataSets (RRID:SCR_005012) under the accession number: GSE89999.

## RT-qPCR for gene expression

Following entrainment (two cycles of entrainment for wild type and the *kaiBC*⁻ strain and one cycle for the *rpaA*⁻ strain) equal ODs of wild type, the *kaiBC*⁻ and the *rpaA*⁻ cells were harvested by filtering and freezing in liquid nitrogen. For the experiment described in *Figure 3—figure supplement 2* cells were harvested at dusk (ZT = 12 hr) and 5 min, 15 min, 30 min, 1 hr and 2 hr following exposure to darkness. For the experiment described in *Figure 4B* cells were harvested in the morning (ZT = 3 hr) and in the afternoon (ZT = 9 hr). RNA extraction was performed as above. RT-qPCR was carried out using SYBRGreen PCR Master Mix (Applied Biosystems), Superscript III Reverse Transcriptase (Invtrogen), and the following primers:

| Target | Primer sequences |
|---|---|
| *gnd* | F: TAGGTGAACTGGCGCGGATT |
| | R: GGGTGCCAGCAACAGATTCG |
| *glgP* | F: TGATAGTCCGCCATCAACAT |
| | R: GATCGCTTTAGCAGTGGTCA |
| *zwf* | F: CACCGCAGACAAACTGAGAT |
| | R: TGACGGTATCCGTAACGAAA |
| *pgm1 (synpcc7942_0156)* | F: CTCGATCGACTCGGTAGTCA |
| | R: AATGCGATCGAAGTCAAACA |
| *pgm2 (synpcc7942_1268)* | F: GCAGCTGATTTCACCTTTGA |
| | R: TGCTGGCAAATTCTTCTGAC |
| *glgA* | F: TAATCACTTCGCGGTTTACG |
| | R: GCCAGTCTTCGTCTTCTCCT |
| *glgC* | F: AATTGCATCAACGCTGACAT |
| | R: CTAGCACCTCAACAAAGCCA |
| *bub2 (S. cerevisiae)* | F: CCTTCCACAACCATTTACCA |
| | R: AAGCAAAGCACGACAGACAC |

The abundance of transcripts was normalized by an external spike-in *bub2* transcript from *S. cerevisiae*. *bub2* was in vitro transcribed, added to the RNA AE extraction buffer and extracted together with *S. elongatus* RNA.

## Enzyme activity assays

All enzyme assays were performed using fresh cultures harvested immediately before the assay. Cells were collected by centrifugation. To prepare lysates cells were resuspended in an appropriate assay buffer as described below with Complete EDTA-Free Protease Inhibitors (Roche), transferred to 2 ml screw-cap tubes containing 0.1 mm glass beads (Research Products International Corp) and lysed by ten 30 s long cycles of bead-beating at 4°C with periodic cooling on ice. Protein concentration of each sample was determined by Bradford assay (Bio-Rad). Assays were performed in 96-well plate in the reaction volume of 200 µl. Glycogen phosphorylase activity was assayed in a buffer containing 18 mM $KH_2PO_4$, 27 mM $Na_2HPO_4$, 15 mM $MgCl_2$ and 100 µM EDTA with 340 µM $Na_2NADP^+$, 4 µM glucose-1,6-bisphosphate (Sigma-Aldrich), 0.8 U $ml^{-1}$ phosphoglucomutase (Sigma-Aldrich), 6 U $ml^{-1}$ glucose-6-phosphate dehydrogenase (Sigma-Aldrich) and 2 mg $ml^{-1}$ glycogen from bovine liver (Sigma-Aldrich) as substrate (*Fu and Xu, 2006*). Glucose-6-phosphate dehydrogenase was assayed in 10 mM $MgCl_2$ and 50 mM Tris maleate, pH 7.5 with 2 mM $NADP^+$ and 4 mM glucose-6-phosphate (Sigma-Aldrich) as substrate (*Schaeffer and Stanier, 1978*). 6-phosphogluconate dehydrogenase was assayed in 10 mM $MgCl_2$ and 50 mM Tris maleate, pH 7.5 with 2 mM $NADP^+$ and 2 mM 6-phosphogluconate (Sigma-Aldrich) as substrate (*Schaeffer and Stanier, 1978*). Phosphoglucomutase was assayed in 50 mM Tris-HCl, pH 8.0 and 10 mM DTT with 2 mM $MgCl_2$, 10 µM glucose-1,6-bisphosphate, 1 mM $NADP^+$, 0.2 U $ml^{-1}$ glucose-6-phosphate dehydrogenase and 2 mM glucose-1-phosphate (Sigma-Aldrich) as substrate (*Liu et al., 2013*). Glycogen synthase was assayed in a three-step reaction following *Suzuki et al. (2010)*. First, ADP was generated from ADP-glucose in a reaction consisting of 50 mM Tris-HCl, pH 8.0, 20 mM DTT, 2 mM ADP-glucose (Sigma-Aldrich), 2 mg $ml^{-1}$ oyster glycogen (Sigma-Aldrich) as substrate and the cell extract. The reaction was incubated for 20 min at 30°C and then stopped by heat treatment at 100°C for 2 min. Then, ATP was generated from ADP by mixing the solution 3:1 with a buffer containing 50 mM HEPES-NaOH, pH 7.5, 10 mM phosphocreatine (Sigma-Aldrich), 200 mM KCl, 10 mM $MgCl_2$, and 0.5 mg $ml^{-1}$ creatine phosphokinase (Sigma-Aldrich), and incubated for 30 min at 30°C. The reaction was stopped by heat treatment in a 100°C metal block for 2 min. Finally, the amount of ATP was measured by mixing the solution 3:2 with a buffer comprising of 125 mM HEPES-NaOH, pH 7.5, 10 mM glucose, 20 mM $MgCl_2$, and 1 mM $NADP^+$ and 5 U $ml^{-1}$ each of hexokinase (Sigma-Aldrich) and glucose-6-phosphate dehydrogenase. All above assays were measured spectrophotometrically by monitoring the increase in absorbance at 340 nm over time using Spectramax i3 plate reader. ADP-glucose pyrophosphorylase was assayed in 50 mM HEPES, pH 8.0, 10 mM $MgCl_2$ with 2 mM ATP, 2 mM 3-phosphoglycerate (Santa Cruz Biotechnology), 1 U $ml^{-1}$ yeast inorganic pyrophosphatase (Sigma-Aldrich) and 1 mM glucose-1-phosphate (Sigma-Aldrich) as substrate (*Díaz-Troya et al., 2014*). Production of phosphate was monitored using EnzCheck Assay (Molecular Probes) by reading absorbance at 360 nm over time in Spectramax i3 plate reader. To determine background control assays for each enzymatic activity were performed without addition of the relevant substrate.

## Glycogen measurements

Glycogen measurements were performed following *Pattanayak et al. (2015)* with modifications. Briefly, for each time point 10 ml of cultures were collected by filtering and freezing in liquid nitrogen. Cells were resuspended from the filter in 1 ml ice-cold methanol. Filters were discarded and cells were centrifuged at 14000 rpm for 7 min. The chlorophyll content was determined spectrophotometrically at 665 nm using the equation $C_{Chl} = A_{665} \times 13.9$ µg $ml^{-1}$ (*de Marsac and Houmard, 1988*). Glycogen content was normalized by chlorophyll content in each sample. The pellets were resuspended in 300 µl of 40% KOH and incubated for 90 min at 95°C. 200 proof ethanol was added to each extract and extracts were incubated at −20°C overnight. Samples were centrifuged at 14000 rpm for 1 hr at 4°C, the supernatants were discarded and the pellets were washed twice with ice-cold ethanol. The pellets were resuspended in 100 µl of 2 N HCL and placed in a 95°C heat block for 30 min to break glycogen down. Samples were neutralized with 100 µl of 2 N NaOH and 50 µl of 1 M phosphate buffer, pH 7. Glycogen content was then assayed enzymatically using glucose

hexokinase assay (Sigma-Aldrich) in 96-well plate at 340 nm in the Spectramax i3 plate reader. Glycogen from bovine liver (Sigma-Aldrich) was used to generate a standard curve.

## Acknowledgements

We thank Andrew Murray, Joseph Markson, Brian Zid, Alicia Darnell, Xiao-yu Zheng, Christopher Chidley, Lauren Surface, Roarke Kamber and James Martenson for comments on the manuscript. We thank Vladimir Denic and members of the O'Shea and Denic labs for helpful discussions and constructive criticism, and Christian Daly and Jennifer Couget from the Bauer Core Facility for performing sequencing runs and for technical help. We thank Peter Arvidson and Jeffrey Offermann for technical support. We thank Joseph Markson and Susan Golden for plasmids. This work was funded by the Howard Hughes Medical Institute.

## Additional information

### Competing interests

EKO: Chief Scientific Officer and a Vice President at the Howard Hughes Medical Institute, one of the three founding funders of eLife. The other author declares that no competing interests exist.

### Funding

| Funder | Author |
| --- | --- |
| Howard Hughes Medical Institute | Erin K O'Shea |

The funders had no role in study design, data collection and interpretation, or the decision to submit the work for publication.

### Author contributions

AMP, Conceptualization, Formal analysis, Investigation, Writing—original draft, Writing—review and editing; EKO, Conceptualization, Formal analysis, Writing—original draft, Writing—review and editing

### Author ORCIDs

Anna M Puszynska, http://orcid.org/0000-0002-8751-5335
Erin K O'Shea, http://orcid.org/0000-0002-2649-1018

## Additional files

### Supplementary files

• Supplementary file 1. Genes whose expression is abrogated in the *rpaA*⁻ 'clock rescue' strain.

• Supplementary file 2. Strain list. Cmr, chloramphenicol resistance; Gmr, gentamycin resistance; Kmr, kanamycin resistance; Ntr, nourseothricin resistance; Sp/St, spectinomycin/streptomycin resistance; NS 1, neutral site 1 (GenBank U30252)

### Major datasets

The following dataset was generated:

| Author(s) | Year | Dataset title | Dataset URL | Database, license, and accessibility information |
| --- | --- | --- | --- | --- |
| Puszynska AM, O'Shea EK | 2017 | Analysis of gene expression in the "clock rescue" and rpaA- "clock rescue" strains of Synechococcus elongatus PCC7942 with RNA sequencing in light/dark conditions | https://www.ncbi.nlm.nih.gov/geo/query/acc.cgi?acc=GSE89999 | Publicly available at the NCBI Gene Expression Omnibus (accession no. GSE89999) |

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
