## [Decision Letter]

Thank you for submitting your article "Switching of metabolic programs in response to light availability is an essential role of cyanobacterial circadian clock" for consideration by *eLife*. Your article has been favorably evaluated by Naama Barkai (Senior Editor) and three reviewers, one of whom, Michael Laub, is a member of our Board of Reviewing Editors. The following individuals involved in review of your submission have agreed to reveal their identity: Michael J Rust (Reviewer #2); Andy LiWang (Reviewer #3).

The reviewers have discussed the reviews with one another and the Reviewing Editor has drafted this decision to help you prepare a revised submission.

Summary:

The reviewers and editor all agreed that this paper presents an important advance in the study of cyanobacterial circadian rhythms. Of the many genes and pathways that are circadian regulated, it has been unclear which are required for viability in conditions of cycling light and dark. These authors have now nicely pinpointed glycogen storage and carbon metabolism as a critical target. The reviewers and editor discussed the manuscript and have only a few issues that should be addressed in a revised version of the manuscript, summarized below.

Essential revisions:

The presentation in the title and abstract focuses largely on clock regulation of metabolism and a general audience that skims the paper might come away with the impression that the study shows an essential function for the clock in LD cycles. The authors show that the transcription factor RpaA is essential in light-dark cycles and have carried out a thorough analysis of its essential role. But it is important to logically separate the essential need for phosphorylated RpaA from the clock control of RpaA phosphorylation.

There may well be other, clock-independent factors that influence RpaA activity or the RpaA regulon (e.g. direct redox sensing, a role for RpaB, the histidine kinase CikB recently described by the Golden lab), and it may be that the loss of these other inputs is more important for viability than the circadian clock per se. It is worth noting along these lines that *kai* gene mutants typically have much less severe phenotypes than *rpaA*-null. Also, the original screen by Ashby & Mullineaux identified *rpaA* (regulator of phycobilisome association) not through a defect in cycling conditions, but as a mutant with constitutively abnormal photosynthetic properties. So, I think the Discussion should be tempered along these lines.

The following three experimental suggestions may be worth considering to resolve the above two points, though they are not strictly required for a revised manuscript to be reconsidered:

1) Does the RpaD53E phosphomimic previously reported by this lab show growth defects (presumably this mutant lacks both clock control of transcription and any other hypothetical control of RpaA by histidine kinases/phosphatases)?

2) Does the *rpaA*-dependent induction of key metabolic genes such as *glgP* and *gap1* in the hours immediately following lights-off still occur in a clock-null strain? That is, is a signal from the clock needed to express metabolic genes at dusk?

3) Determine whether the phosphorylation status of RpaA changes during light-dark cycles in a *kai*-null mutant.

The mechanism by which RpaA regulates glycogen metabolism remains mysterious, because 1) levels and activities of glycogen-synthetic enzymes in *rpaA*^-^ strains were commensurate to those in wild type cyanobacteria, and 2) in *rpaA*^-^ strains with IPTG-inducible enzymes and the ability to import exogenous glucose, glycogen stores remained low even with high cellular energy charge. Perhaps *rpaA*^-^ cells breakdown glycogen too rapidly? Ideally the authors would experimentally address this possibility and other alternatives that would provide a mechanistic understanding of the observations reported. At a minimum, the authors should discuss these alternatives in the paper.

---

## [Author Response]

*Essential revisions:*

*The presentation in the title and abstract focuses largely on clock regulation of metabolism and a general audience that skims the paper might come away with the impression that the study shows an essential function for the clock in LD cycles. The authors show that the transcription factor RpaA is essential in light-dark cycles and have carried out a thorough analysis of its essential role. But it is important to logically separate the essential need for phosphorylated RpaA from the clock control of RpaA phosphorylation.*

*There may well be other, clock-independent factors that influence RpaA activity or the RpaA regulon (e.g. direct redox sensing, a role for RpaB, the histidine kinase CikB recently described by the Golden lab), and it may be that the loss of these other inputs is more important for viability than the circadian clock per se. It is worth noting along these lines that kai gene mutants typically have much less severe phenotypes than rpaA-null. Also, the original screen by Ashby & Mullineaux identified rpaA (regulator of phycobilisome association) not through a defect in cycling conditions, but as a mutant with constitutively abnormal photosynthetic properties. So, I think the Discussion should be tempered along these lines.*

*The following three experimental suggestions may be worth considering to resolve the above two points, though they are not strictly required for a revised manuscript to be reconsidered:*

1) Does the RpaD53E phosphomimic previously reported by this lab show growth defects (presumably this mutant lacks both clock control of transcription and any other hypothetical control of RpaA by histidine kinases/phosphatases)?

We thank the reviewers for bringing to our attention this logical incoherence. Based on the results of our new experiments, we propose that the phosphorylated species of RpaA is crucial for surviving light/dark cycles, and that main regulation of the phosphorylation state of RpaA in light/dark cycling conditions is clock-phosphorylation dependent. We conclude that any additional non-circadian pathways that may modulate RpaA activity (and, by extension, cell viability) do not play a significant role under these conditions. We first investigated which phosphorylated form of RpaA was important for surviving light/dark conditions and find that RpaA D53E, mimicking the phosphorylated state, is sufficient to rescue growth, whereas RpaA D53A that mimics the unphosphorylated state, is not (Figure 1). While this result confirms the key role of RpaA phosphorylation for cell viability under light/dark conditions, it does not dissect the contributions of the core clock and the potential non-circadian factors to regulation of RpaA activity. To address this issue, we analyzed RpaA phosphorylation and cell viability in mutants expressing phosphomimetic variants of KaiC, KaiC-AA (KaiC S431A T432A) and KaiC-EE (KaiC S431E T432E), corresponding to the unphosphorylated and the hyperphosphorylated states of KaiC, respectively (Figure 1). We hypothesized that these KaiC variants would have different effects on the activity of the downstream histidine kinases, CikA and SasA and, hence, on RpaA phosphorylation. To ensure unperturbed expression of KaiC phosphomimetics, we replaced the *kaiBC* promoter with an RpaA-independent promoter. We observed that the _c_KaiC-AA strain (subscript “c” denoting constitutive, RpaA-independent expression), which exhibits an elevated RpaA phosphorylation level, is able to grow in light/dark conditions whereas the _c_KaiC-EE strain, in which RpaA phosphorylation is low, is inviable in these conditions. Assuming that the alternative factors influencing RpaA activity (direct redox sensing, RpaB, CikB) act independently of the clock and that their function is unaffected by point mutations in KaiC, this result suggests that regulation of RpaA phosphorylation state in diurnal conditions is dependent on the clock phosphorylation state.

While we propose that circadian control of RpaA activity fully accounts for the loss of fitness of the *rpaA^-^*strain in the light/dark conditions we studied, it is possible that alternative regulators of RpaA (direct redox sensing, RpaB, CikB) play a significant role in preserving cell viability under other conditions. For example, it is possible that in an environment in which light intensity fluctuates during the day, the hypothetical ability of RpaA to integrate environmental signals through the action of additional factors is key for cell fitness. We have added this point to our Discussion.

*2) Does the rpaA-dependent induction of key metabolic genes such as glgP and gap1 in the hours immediately following lights-off still occur in a clock-null strain? That is, is a signal from the clock needed to express metabolic genes at dusk?*

We measured expression of the key metabolic genes in the *kaiBC^-^*strain (Figure 3—figure supplement 2) and we observe that they are expressed at a similar level in wild type cells and in the *kaiBC^-^*strain. In the *kaiBC^-^*strain, RpaA phosphorylation is maintained at a high level throughout the light/dark cycle allowing for expression of dusk-specific genes in this strain.

Interestingly, Hosokawa et al. demonstrated that expression profiles of a small subset of dark-induced genes are changed in the *kaiBC^-^*strain compared to wild type during darkness (Hosokawa et al., 2011). These changes may be due to the fact that in the *kaiBC^-^*strain RpaA phosphorylation state is kept at a constant level lacking diurnal dynamics, and is also somewhat lower than the maximum level of RpaA phosphorylation observed in wild type.

*3) Determine whether the phosphorylation status of RpaA changes during light-dark cycles in a kai-null mutant.*

Complexes of KaiB and KaiC are required in vitro to stimulate the phosphatase activity of CikA (Gutu and O’Shea, 2013). Therefore, we expected that the *kaiBC^-^*strain grown in light/dark conditions would maintain RpaA phosphorylation at an elevated level due to the loss of stimulation of CikA activity by KaiB-KaiC complexes. We performed Phos-tag westerns in the *kaiBC^-^*strain and we indeed observed that RpaA phosphorylation remains high and constant throughout the light/dark cycle (Figure 1). The presence of high levels of phosphorylated RpaA in the *kaiBC^-^* strain provides an explanation for this strain’s lack of a significant growth defect in light/dark conditions. This result and our observation that the phosphomimetic version of RpaA (RpaA D53E) can complement the viability defect of the *rpaA^-^*strain are consistent with the model that phosphorylated RpaA is the species of RpaA required to survive light/dark cycles.

*The mechanism by which RpaA regulates glycogen metabolism remains mysterious, because 1) levels and activities of glycogen-synthetic enzymes in rpaA- strains were commensurate to those in wild type cyanobacteria, and 2) in rpaA- strains with IPTG-inducible enzymes and the ability to import exogenous glucose, glycogen stores remained low even with high cellular energy charge. Perhaps rpaA^-^ cells breakdown glycogen too rapidly? Ideally the authors would experimentally address this possibility and other alternatives that would provide a mechanistic understanding of the observations reported. At a minimum, the authors should discuss these alternatives in the paper.*

We thank the reviewers for this suggestion and we have included it in our Discussion. While we agree that it is formally possible that the *rpaA^-^*cells perform a futile cycle synthesizing and degrading glycogen simultaneously, we think that this scenario is not likely given that the expression level and activity of glycogen phosphorylase required for glycogen breakdown are very low in the *rpaA^-^*strain. Additionally, we analyzed the expression of glycogen debranching enzymes present in *S. elongatus (synpcc7942_0086* and *synpcc7942_1575*) using our RNA-seq expression data, and we found that expression of each of them is lower in the *rpaA^-^*‘clock rescue’ strain than in the control (data not shown). We propose that the ability to degrade glycogen is defective in the *rpaA^-^*strain.

Since glucose supplementation does not increase glycogen stores in wild type cells expressing GalP (we added the data as Figure 5—figure supplement 2), we did not expect glucose treatment to increase glycogen synthesis in the *rpaA^-^*strains with IPTG-inducible enzymes and the ability to import exogenous glucose. Instead, we believe that supplementation with glucose provides an external carbon source supplanting the need for formation of internal carbon stores in the form of glycogen.